# ONLINE LIMITED MEMORY NEURAL-LINEAR BANDITS

## ABSTRACT

We study neural-linear bandits for solving problems where *both* exploration and representation learning play an important role. Neural-linear bandits leverage the representation power of deep neural networks and combine it with efficient exploration mechanisms, designed for linear contextual bandits, on top of the last hidden layer. Since the representation is optimized during learning, information regarding exploration with "old" features is lost. We propose the first limited memory neural-linear bandit that is resilient to this catastrophic forgetting phenomenon by solving a semi-definite program. We then approximate the semi-definite program using stochastic gradient descent to make the algorithm practical and adjusted for online usage. We perform simulations on a variety of data sets, including regression, classification, and sentiment analysis. We observe that our algorithm achieves superior performance and shows resilience to catastrophic forgetting.

## 1 INTRODUCTION

Deep neural networks (DNNs) can learn representations of data with multiple levels of abstraction and have dramatically improved the state-of-the-art in speech recognition, visual object recognition, object detection, and many other domains such as drug discovery and genomics (LeCun et al., 2015; Goodfellow et al., 2016). Using DNNs for function approximation in reinforcement learning (RL) enables the agent to generalize across states without domain-specific knowledge, and learn rich domain representations from raw, high-dimensional inputs (Mnih et al., 2015; Silver et al., 2016).

Nevertheless, the question of how to perform efficient exploration during the representation learning phase is still an open problem. The $\epsilon$-greedy policy (Langford & Zhang, 2008) is simple to implement and widely used in practice (Mnih et al., 2015). However, it is statistically suboptimal. Optimism in the Face of Uncertainty (Abbasi-Yadkori et al., 2011; Auer, 2002, OFU), and Thompson Sampling (Thompson, 1933; Agrawal & Goyal, 2013, TS) use confidence sets to balance exploitation and exploration. For DNNs, such confidence sets may not be accurate enough to allow efficient exploration. For example, using dropout as a posterior approximation for exploration does not concentrate on observed data (Osband et al., 2018) and was shown empirically to be insufficient (Riquelme et al., 2018). Alternatively, pseudo-counts, a generalization of the number of visits, were used as an exploration bonus (Bellemare et al., 2016; Pathak et al., 2017). Inspired by tabular RL, these ideas ignore the uncertainty in the value function approximation in each context. As a result, they may lead to inefficient confidence sets (Osband et al., 2018).

Linear models, on the other hand, are considered more stable and provide accurate uncertainty estimates but require substantial feature engineering to achieve good results. Additionally, they are known to work in practice only with "medium-sized" inputs (with around $1,000$ features) due to numerical issues. A natural attempt at getting the best of both worlds is to learn a linear exploration policy on top of the last hidden layer of a DNN, which we term the **neural-linear** approach. In RL, this approach was shown to refine the performance of DQNs (Levine et al., 2017) and improve exploration when combined with TS (Azizzadenesheli et al., 2018) and OFU (O'Donoghue et al., 2018; Zahavy et al., 2018a). For contextual bandits, Riquelme et al. (2018) showed that neural-linear TS achieves superior performance on multiple data sets.

A practical challenge for neural-linear bandits is that the representation (the activations of the last hidden layer) change after every optimization step, while the features are assumed to be fixed over time when used by linear contextual bandits. Zhou et al. (2019) recently suggested to analyze deep contextual bandits with an "infinite width" via the Neural Tangent Kernel (NTK) (Jacot et al., 2018). Under the NTK assumptions, the optimal solution (and its features) are guaranteed to be close to the initialization point, so that the deep bandit can be viewed as a kernel method. Riquelme et al. (2018), on the other hand, observed that with standard DNN architectures, the features do change from the initialization point and a mechanism to adapt for that change is required. They tackled this problem by storing the entire data set in a memory buffer and computing new features for all the data

after each DNN learning phase. The authors also experimented with a bounded memory buffer but observed a significant decrease in performance due to **catastrophic forgetting** (Kirkpatrick et al., 2017), i.e., a loss of information from previous experience.

In this work, we propose a neural-linear bandit that uses TS on top of the last layer of a DNN. Key to our approach is a novel method to compute priors whenever the DNN features change that makes our algorithm resilient to catastrophic forgetting. Specifically, we adjust the moments of the likelihood of the reward estimation conditioned on new features to match the likelihood conditioned on old features. We achieve this by solving a semi-definite program (Vandenberghe & Boyd, 1996, SDP) to approximate the covariance and using the weights of the last layer as prior to the mean. To make the algorithm more appealing for real-time usage, we implement it in an online manner, in which updates of the DNN weights and the priors are done simultaneously every step by using stochastic gradient descent (SGD) followed by projection of the priors. This obviates the need to process the whole memory buffer after each DNN learning phase and keeps the computational burden of our algorithm small.

We performed experiments on several real-world and simulated data sets, including classification and regression, using Multi-Layered Perceptrons (MLPs). These experiments suggest that our prior approximation scheme improves performance significantly when memory is limited. We demonstrate that our neural-linear bandit performs well in a sentiment analysis data set where the input is given in natural language (there are $8k$ features), and we use a Convolution Neural Network (CNNs). In this regime, it is not feasible to use a linear method due to computational problems. In addition, we evaluate our algorithm in a stochastic simulation of an uplink video-transmission application. In this application, the length of the simulation is so long that it is not possible to use the unlimited memory neural-linear approach of Riquelme et al. (2018). To the best of our knowledge, this is the first neural-linear algorithm that is resilient to catastrophic forgetting due to limited memory. In addition, unlike Riquelme et al. (2018), which use a patch-based approach, our algorithm can be configured to work in an online manner, in which the DNN and statistics are efficiently updated each step. Thus, this is also the first neural-linear online algorithm.

## 2 BACKGROUND

**The stochastic, contextual (linear) multi-armed bandit problem.** At every time $t$, a contextual bandit algorithm observes a context $b(t)$ and chooses an arm $a(t) \in [1, \ldots, N]$. The bandit can use the history $H_{t-1}$ to make its decisions, where $H_{t-1} = \{b(\tau), a(\tau), r_{a(\tau)}(\tau), \tau = 1, ..., t-1\}$, and $a(\tau)$ denotes the arm played at time $\tau$. Most existing works typically make the following **realizability** assumption (Chu et al., 2011; Abbasi-Yadkori et al., 2011; Agrawal & Goyal, 2013).

**Assumption 1.** *The reward for arm $i$ at time $t$ is generated from an (unknown) distribution s.t.* $\mathbb{E}[r_i(t)|b(t), H_{t-1}] = \mathbb{E}[r_i(t)|b(t)] = b(t)^T \mu_i$, *where* $\{\mu_i \in \mathbb{R}^d\}_{i=1}^N$ *are fixed but unknown.*

Let $a^*(t)$ denote the optimal arm at time t, i.e. $a^*(t) = \arg\max_i b(t)^T \mu_i$, and let $\Delta_i(t)$ the difference between the mean rewards of the optimal arm and of arm $i$ at time $t$, i.e., $\Delta_i(t) = b(t)^T \mu_{a^*(t)} - b(t)^T \mu_i$. The objective is to minimize the total regret $R(T) = \sum_{t=1}^T \Delta_{a(t)}$, where $T$ is finite.

**TS for linear contextual bandits.** Thompson sampling is an algorithm for online decision problems where actions are taken sequentially in a manner that must balance between exploiting what is known to maximize immediate performance and investing to accumulate new information that may improve future performance (Russo et al., 2018; Lattimore & Szepesvári, 2018). For linear contextual bandits, TS was introduced in (Agrawal & Goyal, 2013, Alg. 1).

Suppose that the **likelihood** of reward $r_i(t)$, given context $b(t)$ and parameter $\mu_i$, were given

---

**Algorithm 1** TS for linear contextual bandits

$\forall i \in [1.., N]$, set $\Phi_i = I_d$, $\hat{\mu}_i = 0_d$, $f_i = 0_d$
**for** $t = 1, 2, \ldots,$ **do**
   $\forall i \in [1.., N]$, sample $\tilde{\mu}_i$ from $N(\hat{\mu}_i, v^2 \Phi_i^{-1})$
   Play arm $a(t) := \arg\max_i b(t)^T \tilde{\mu}_i$
   Observe reward $r_t$
   **Update:** $\Phi_{a(t)} = \Phi_{a(t)} + b(t)b(t)^T$
   $f_{a(t)} = f_{a(t)} + b(t)r_t$,   $\hat{\mu}_{a(t)} = \Phi_{a(t)}^{-1} f_{a(t)}$
**end for**

---

by the pdf of Gaussian distribution $N(b(t)^T \mu_i, \nu^2)$, and let $\Phi_i(t) = \Phi_i^0 + \sum_{\tau=1}^{t-1} b(\tau)b(\tau)^T \mathbb{1}_{i=a(\tau)}$, $\hat{\mu}_i(t) = \Phi_i^{-1}(t) \sum_{\tau=1}^{t-1} b(\tau)r_{a(\tau)}(\tau) \mathbb{1}_{i=a(\tau)}$, where $\mathbb{1}$ is the indicator function and $\Phi_i^0$ is the precision prior. Given a Gaussian **prior** for arm $i$ at time $t$, $N(\hat{\mu}_i(t), v^2 \Phi_i^{-1}(t))$, the **posterior** distribution at time $t+1$ is given by,

$$Pr(\tilde{\mu}_i|r_i(t)) \propto Pr(r_i(t)|\tilde{\mu}_i)Pr(\tilde{\mu}_i) \propto N(\hat{\mu}_i(t+1), v^2 \Phi_i^{-1}(t+1)).$$

At each time step $t$, the algorithm generates samples $\{\tilde{\mu}_i(t)\}_{i=1}^N$ from the posterior distribution $N(\hat{\mu}_i(t), v^2\Phi_i^{-1}(t))$, plays the arm $i$ that maximizes $b(t)^T\tilde{\mu}_i(t)$ and updates the posterior. TS is guaranteed to have a total regret at time $T$ that is not larger than $O(d^{3/2}\sqrt{T})$, which is within a factor of $\sqrt{d}$ of the information-theoretic lower bound for this problem. It is also known to achieve excellent empirical results (Lattimore & Szepesvári, 2018). Although that TS is a Bayesian approach, the description of the algorithm and its analysis are prior-free, i.e., the regret bounds will hold irrespective of whether or not the actual reward distribution matches the Gaussian likelihood function used to derive this method (Agrawal & Goyal, 2013).

## 3 Limited memory neural-linear TS

**Algorithm.** Our algorithm is composed of four main components: **(1) Representation:** A DNN takes the raw context as an input and is trained to predict the reward of each arm; **(2) Exploration:** a mechanism that uses the last layer activations of the DNN as features and performs linear TS on top of them; **(3) Memory** a buffer that stores previous experience; **(4) Likelihood matching:** a mechanism that uses the memory buffer and the DNN to account for changes in representation.

To derive our algorithm, we make a **realizability assumption**, which is similar to Assumption 1. The difference is that we assume that **all** the representations that are produced by the DNN are **realizable**.

**Assumption 2.** *For any representation $\phi$ that is produced by the DNN, the reward for arm $i$ at time $t$ is generated from an (unknown) distribution s.t. $\mathbb{E}\left[r_i(t)|\phi(t), H_{t-1}\right] = \mathbb{E}\left[r_i(t)|\phi(t)\right] = \phi(t)^T\mu_i$, where $\{\mu_i \in \mathbb{R}^d\}_{i=1}^N$ are fixed but unknown parameters.*

That is, for each representation there exist a *different* linear coefficients vector (e.g. $\mu$ for $\phi$, $\beta$ for $\psi$,) such that the expected reward is linear in the features. While this assumption may be too strong to hold in practice, it allows us to derive our algorithm as a good approximation that performs extremely well on many problems. We now explain how each of these components works; **code** can be found in (link), and **pseudo code** in the supplementary (Algorithm 2).

**1. Representation.** Our algorithm uses a DNN, denoted by $D_\omega$, where $\omega$ denotes the DNN's weights (for convenience, we exclude the weight notation for the rest of the paper). The DNN takes the raw context $b(t) \in \mathbb{R}^d$ as its input. The network has $N$ outputs that correspond to the estimation of the reward of each arm, given context $b(t) \in \mathbb{R}^d$, $D(b(t))_i$ denotes the estimation of the reward of the $i$-th arm.

Using a DNN to predict each arm's reward allows our algorithm to learn a nonlinear representation of the context. This representation is later used for exploration by performing linear TS on top of the last hidden layer activations. We denote the activations of the last hidden layer of $D$ applied to this context as $\phi(t) = \text{LastLayerActivations}(D(b(t)))$, where $\phi(t) \in \mathbb{R}^g$. The context $b(t)$ represents raw measurements that can be high dimensional (e.g., image or text), where the size of $\phi(t)$ is a design parameter that we choose to be smaller ($g < d$). This makes contextual bandit algorithms practical for such data sets. Moreover, $\phi(t)$ can potentially be linearly realizable (even if $b(t)$ is not) since a DNN is a global function approximator (Barron, 1993) and the last layer is linear.

**1.1 Training.** Every $L$ iterations, we train $D$ for $P$ mini-batches. Training is performed by sampling experience tuples $\{b(\tau), a(\tau), r_{a(\tau)}(\tau)\}$ from the replay buffer $E$ (details below) and minimizing the mean squared error (MSE),

$$\mathcal{L}_{NN} = ||D(b(\tau))_{a(\tau)} - r_{a(\tau)}(\tau)||_2^2, \tag{1}$$

where $r_{a(\tau)}$ is the reward that was received at time $\tau$ after playing arm $a(\tau)$ and observing context $b(\tau)$. Notice that only the output of arm $a(\tau)$ is differentiated, and that the DNN (including the last layer) is trained end-to-end to minimize Eq. (1).

**2. Exploration.** Since our algorithm is performing training in phases (every $L$ steps), exploration is performed using a fixed representation $\phi$ ($D$ has fixed weights between training phases). At each time step $t$, the agent observes a raw context $b(t)$ and uses the DNN $D$ to produces a feature vector $\phi(t)$. The features $\phi(t)$ are used to perform linear TS, similar to Algorithm 1, but with two key differences: (1) We introduce a likelihood matching mechanism that accounts for changes in representation (2) Instead of using a Gaussian posterior, we use the Bayesian Linear Regression (BLR) formulation that was suggested in Riquelme et al. (2018). Empirically, this update scheme was shown to convergence to the true posterior and demonstrated excellent empirical performance (Riquelme et al., 2018).

In BLR, the noise parameter $\nu$ (Alg. 1) is replaced with a prior belief that is being updated over time. The **prior** for arm $i$ at time $t$ is given by $Pr(\tilde{\mu}_i, \tilde{\nu}_i^2) = Pr(\tilde{\nu}_i^2)Pr(\tilde{\mu}_i|\tilde{\nu}_i^2)$, where $Pr(\tilde{\nu}_i^2)$ is an inverse-gamma distribution Inv-Gamma$(a_i(t), b_i(t))$, and the conditional prior density $Pr(\tilde{\mu}_i|\tilde{\nu}_i^2)$ is a normal

distribution, $Pr(\tilde{\mu}_i|\tilde{\nu}_i^2) \propto \mathcal{N}\left(\hat{\mu}_i(t), \tilde{\nu}_i^2 \Phi_i(t)^{-1}\right)$. Combining this prior with a Gaussian likelihood guarantees that the the **posterior** distribution at time $\tau = t + 1$ is given in the same form (a conjugate prior), i.e., $\Pr(\tilde{\nu}_i) = \text{Inv-Gamma}\left(A_i(\tau), B_i(\tau)\right)$ and $\Pr(\tilde{\mu}_i|\tilde{\nu}_i) = \mathcal{N}\left(\hat{\mu}_i(\tau), \tilde{\nu}_i^2 \Phi_i(\tau)^{-1}\right)$.

In each step and for each arm $i \in 1..N,$, we sample a noise parameter $\tilde{\nu}_i^2$ from $Pr(\tilde{\nu}_i^2)$ and then sample a weight vector $\tilde{\mu}_i$ from the posterior $N\left(\hat{\mu}_i, \tilde{\nu}_i^2 (\Phi_i^0 + \Phi_i)^{-1}\right)$. Once we sampled a weight vector for each arm, we choose to play arm $a(t) = \arg\max_i \phi(t)^T \tilde{\mu}_i$, and observe reward $r_{a(t)}(t)$. This is followed by a posterior update step:

$$\Phi_{a(t)} = \Phi_{a(t)}^0 + \Phi_{a(t)} + \phi(t)\phi(t)^T, \ \ f_{a(t)} = f_{a(t)} + \phi(t)^T r_t,$$
$$\hat{\mu}_{a(t)} = (\Phi_{a(t)})^{-1}\left(\Phi_{a(t)}^0 \mu_{a(t)}^0 + f_{a(t)}\right), \ \ R_i^2(t) = R_i^2(t-1) + r_i^2$$
$$A_i(t) = A_{a(t)}^0 + \frac{t}{2}, \ \ B_i(t) = B_{a(t)}^0 + \frac{1}{2}\left(R_i^2(t) + (\mu_{a(t)}^0)^{\mathrm{T}}\Phi_{a(t)}^0 \mu_{a(t)}^0 - \hat{\mu}_{a(t)}(t)^{\mathrm{T}}\Phi_{a(t)}(t)\hat{\mu}_{a(t)}(t)\right)$$
$$(2)$$

We note that the exploration mechanism only chooses actions; it **does not change** the DNN's weights.

**3.  Memory.**  After an action $a(t)$ is played at time $t$, we store the experience tuple $\{b(t), a(t), r_{a(t)}(t)\}$ in a finite memory buffer of size $n$ that we denote by $E$. Once $E$ is full, we remove tuples from $E$ in a round robin manner, i.e., we remove the first tuple in $E$ with $a = a(t)$.

**4. Likelihood matching.** Before each learning phase, we evaluate the features of $D$ on the replay buffer. Let $E_i$ be a subset of memory tuples in $E$ at which arm $i$ was played, and let $n_i$ be its size. We denote by $E_{\phi^{old}}^i \in \mathbb{R}^{n_i \times g}$ a matrix whose rows are feature vectors that were played by arm $i$. After a learning phase is complete, we evaluate the new activations on the same replay buffer and denote the equivalent set by $E_\phi^i \in \mathbb{R}^{n_i \times g}$. Our approach is to summarize the knowledge that the algorithm has gained from exploring with the features $\phi^{old}$ into priors on the new features $\Phi_i^0, \mu_i^0$. Once these priors are computed, we restart the linear TS algorithm using the data that is currently available in the replay buffer. For each arm $i$, let $\phi_j^i = (E_\phi^i)_j$ be the j-th row in $E_\phi^i$ and let $r_j$ be the corresponding reward, we set $\Phi_i = \sum_{j=1}^{n_i} \phi_j^i(\phi_j^i)^T, f_i = \sum_{j=1}^{n_i} (\phi_j^i)^T r_j$.

We now explain how we compute $\Phi_i^0, \mu_i^0$. Recall that under the realizability assumption we have that $\mathbb{E}[r_i(t)|\phi(t)] = \phi(t)^T \mu_i = \phi^{old}(t)^T \mu_i^{old} = \mathbb{E}[r_i(t)|\psi(t)]$. Thus, the likelihood of the reward is invariant to the choice of representation , i.e. $N(\phi(t)^T \mu_i, \nu^2) \sim N(\phi^{old}(t)^T \mu_i^{old}, \nu^2)$. For all $i$, define the estimator of the reward as $\theta_i(t) = \phi(t)^T \tilde{\mu}_i(t)$, and its standard deviation $s_{t,i} = \sqrt{\phi(t)^T \Phi_i(t)^{-1} \phi(t)}$ (see Agrawal & Goyal (2013) for derivation). By definition of $\tilde{\mu}_i(t)$, marginal distribution of each $\theta_i(t)$ is Gaussian with mean $\phi_i(t)^T \hat{\mu}_i(t)$ and standard deviation $\nu_i s_{t,i}$. The goal is to match the likelihood of the reward estimation $\theta_i(t)$ given the new features to be the same as with the old features.

**4.1 Approximation of the mean $\mu_i^0$:** The DNN is trained to minimize the MSE (Eq. (1)). Given the new features $\phi$, the current weights of the last layer of the DNN already make a good prior for $\mu_i^0$. In Levine et al. (2017), this approach was shown empirically to improve the performance of a neural linear DQN. The main advantage is that the DNN is optimized online by observing all the data and is therefore not limited to the current replay buffer. Thus, the weights of the current DNN hold information on more data and make a strong prior.

**4.2 Approximation of the correlation matrix $\Phi_i^0$:** For each arm $i$, our algorithm receives as input the sets of new and old features $E_\phi^i, E_{\phi^{old}}^i$ with elements $\{\phi_j^{old}, \phi_j\}_{j=1}^{n_i}$. In addition, the algorithm receives the correlation matrix $\Phi_i^{old}$. Notice that due to our algorithm's nature, $\Phi_i^{old}$ holds information on contexts that are not available in the replay buffer. The goal is to find a correlation matrix, $\Phi_i^0$, for the new features that will have the same variance on past contexts as $\Phi_i^{old}$. I.e., we want to find $\Phi_i^0$ such that $\forall i \in [1..N], j \in [1..n_i]$ $s_{j,i}^2 \doteq (\phi_j^{old})^T (\Phi_i^{old})^{-1} \phi_j^{old} = \phi_j^T (\Phi_i^0)^{-1} \phi_j = \text{Trace}\left((\Phi_i^0)^{-1} \phi_j \phi_j^T\right)$, where the last equality follows from the cyclic property of the trace.

We denote by $X_i$ a vector of size $n_i$ in the vector space of $d \times d$ symetric matrices, with its j-th element $X_{j,i}$ to be the matrix $\phi_j \phi_j^T$. Using this notation, we have that Note that $\text{Trace}\left((\Phi_i^0)^{-1}\phi_j\phi_j^T\right) = \text{Trace}(X_{j,i}^T(\Phi_i^0)^{-1})$ is an inner product over the vector space of symmetric matrices, known as the Frobenius inner product. Finally, as $(\Phi_i^0)^{-1}$ is an inverse correlation matrix, we constrain the solution to be a semi positive definite. Thus, the optimization problem is equivalent to a linear regression

problem in the vector space of positive semi definite (PSD) matrices for all actions $i \in [1..N]$ :

$$\underset{(\Phi_i^0)^{-1}}{\text{minimize}} \sum_{j=1}^{n_i} ||\text{Trace}(X_{j,i}^T (\Phi_i^0)^{-1}) - s_{j,i}^2||^2 \quad \text{subject to} \quad (\Phi_i^0)^{-1} \succeq 0. \tag{3}$$

In practice, we solve the SDP by applying SGD using sampled batches from $E_{\phi^{old}}^i$ and $E_{\phi}^i$. Each SGD iteration is followed by eigenvalues thresholding (denoted by EigenValueThresholding($(\Phi_i^0)^{-1}$)) in order to project $(\Phi_i^0)^{-1}$ back to PSD matrices space. To avoid evaluating $E_{\phi}^i$ each time the DNN is updated, we take advantage of the iterative learning phase of the DNN and the iterative nature of the SGD by using the same batch to update the DNN weights and $(\Phi_i^0)^{-1}$ simultaneously. In each iteration, we treat the inverse correlation matrix from the previous iteration as $(\Phi_i^{old})^{-1}$ and also as the initial guess for the current gradient decent step. For each action $a \in A$, we use a subset of the batch, in which action $a$ was used.

**Lemma 1.** *The rank of a stochastic gradient of $\sum_{j=1}^{n_i} ||Trace(X_{j,i}^T (\Phi_i^0)^{-1}) - s_{j,i}^2||^2$, given a batch size $B$ is at most $min\{B, g\}$.*

The proof can be found in the supplementary.

**Computational complexity.** We consider the time and memory complexity of the algorithm and their dependence on different parameters of the problem. Recall that the last layer's dimension is $g < d$ where $d$ is the dimension of the raw features, the size of the replay buffer is $n$ and the batch size is $B$. Therefore, each gradient step is $Bg^2$ (matrix-vector multiplications) plus the thresholding operator, which has a time complexity of $O(g^3)$ due to the matrix eigendecomposition. This can be improved in the case of low-rank stochastic gradients ($B < g$) into $O(g^2)$ as suggested in Chen et al. (2014).

Let $T$ be the number of contexts seen by the algorithm. The computational complexity of the full memory approach results is, therefore, $O(T^2)$, and the memory complexity is $O(T)$. This is because it is estimating the TS posterior using the complete data every time the representation changes. On the other hand, the limited memory approach uses only the memory buffer to estimate the posterior and training the network/updating the priors in a batch manner. This gives a memory complexity of $O(1)$ and computational complexity of $O(T)$; Due to the stochastic behavior of our SGD modification, the computational complexity is linear in the batch size and not in $|A|$.

To summarize, our method is more efficient than the full memory baseline in problems with a lot of data (large $T$). Instead of solving an SDP after each update phase (which is computationally prohibitive in general), we apply efficient SGD in parallel to the DNN updates. This is also sample efficient due to the reuse of the same batch for both tasks. By setting the update frequency to 1 ($L = 1$) and the number of iterations to 1 ($P = 1$), our algorithm becomes fully online in the sense it updates the DNN and the statistics each step. We use the fully online configuration in our experiments to show that even under extreme configuration, our algorithm produces competitive results.

## 4 EXPERIMENTS

In this section, we empirically investigate the performance of the proposed algorithm to address the following questions:

1. Can Neural-linear bandits explore efficiently while learning representations under finite memory constraints?
2. Does the moment matching mechanism allows neural-linear bandits to avoid catastrophic forgetting?
3. Can the method be applied in a wide range of problems and across different DNN architectures?

We address these questions by performing experiments on ten real-world data sets, including a high dimensional natural language data on a task of sentiment analysis (all of these data sets are publicly available through the UCI Machine Learning Repository). In the supplementary, we include an additional experiments: **synthetic data**, where we observe that our algorithm can learn nonlinear representations during exploration and **sentiment analysis from text**, in which we evaluate our algorithm on a text-based dataset using CNNs.

**Methods and setup.** We experimented with different ablations of our approach, as well as a few baselines: **(1)** Linear TS (Agrawal & Goyal, 2013, Algorithm 1) using the raw context as a feature, with an additional uncertainty in the variance (Riquelme et al., 2018). **(2)** Neural-Linear TS (Riquelme

et al., 2018). **(3)** Our neural-linear TS algorithm with limited memory. **(4)** An ablative version of (3) that calculates the prior only for the mean, similar to Levine et al. (2017). **(5)** An ablative version of (3) that does not use prior calculations. Algorithms 3-5 make an ablative analysis for the limited memory neural-linear approach. As we will see, adding each one of the priors improves learning and exploration. In all versions of our algorithm, we set $P = 1, L = 1$, which makes it work in an online manner. In all the experiments, we used the same hyperparameters as in Riquelme et al. (2018). E.g., the network architecture is an MLP with a single hidden layer of size 50. The only exception is with the text CNN (details below). The size of the **memory buffer** is set to be 100 per action.

## 4.1 CATASTROPHIC FORGETTING.

We begin with an illustrative example on the Shuttle Statlog dataset (Newman et al., 2008). Each context is composed of 9 features describing the space shuttle flight. The goal is to predict the state of the radiator of the shuttle (the reward). There are $k = 7$ possible actions; for correct predictions the reward is $r = 1$ and $r = 0$ otherwise.

Fig. 1 shows the performance of each of the algorithms in this setup. We let each algorithm run for 4000 steps (contexts) and average each algorithm over 10 runs. The x-axis corresponds to the number of contexts seen so far, while the y-axis measures the instantaneous regret (lower is better). For this experiment, all Neural-Linear methods retrained the DNN every $L = 400$ steps for $P = 800$ mini-batches. The cumulative reward achieved by each algorithm averaged over seeds (mean and std) can be found in Table 5 (Statlog); in Fig. 1 we focus on the qualitative behavior, as described below.

First, we can see that the neural linear method (green) outperforms the linear one (magenta), suggesting that this data set requires a nonlinear function approximation. We can also see that our approach to computing the priors allows the limited memory algorithm (red) to perform almost as good as the neural linear algorithm without memory constraints (green).

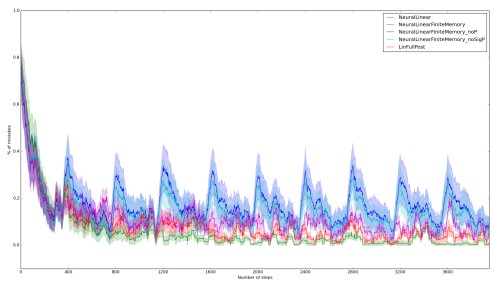

We can also see that the two limited memory neural linear algorithms that do not calculate the prior for the covariance matrix (blue and teal) suffer from "catastrophic forgetting" due to limited memory. Intuitively, the covariance

Figure 1: Catastrophic forgetting

matrix holds information regarding the number of contexts seen by the agent and used by the algorithm for exploration. When no such prior is available, the agent explores sub-optimal arms from scratch every time the features are modified (every $L = 400$ steps, marked by the x-ticks on the graph). Indeed, we observe "peaks" in the regret curve for these algorithms (blue and teal); this is significantly reduced when we compute the prior on the covariance matrix (red), making the limited memory neural-linear bandit resilient to catastrophic forgetting.

## 4.2 REAL WORLD DATA.

We evaluate our approach on ten real-world data sets; for each data set, we present the cumulative reward achieved by each algorithm, averaged over 50 runs. Each run was performed for 5000 steps. The results are divided into two groups, **linear, and nonlinear data sets**. The separation was performed post hoc, based on the results achieved by the full memory methods, i.e., the first group consists of five data sets on which Linear TS (Algorithm 1) outperformed Neural-Linear TS (Algorithm 2), and vice versa. We observed that most of the linear datasets consisted of a small number of features that were mostly categorical (e.g., the mushroom data set has 22 categorical features that become 117 binary features). The DNN based methods performed better when the features were dense and high dimensional.

Since there is no apriori reason to believe that real-world data sets should be linear, we were surprised that the linear method made a competitive baseline to DNNs. To investigate this further, we experimented with the best reported MLP architecture for the covertype data set (taken from Kaggle). Linear methods were reported (link) to achieve around 60% test accuracy. This number is consistent with our reported cumulative reward (3000 out of 5000). Similarly, DNNs achieved around 60% accuracy, which indicates that the Covertype data set is relatively linear. However, when we measure the cumulative reward, the deep methods take initial time to learn, which can explain the slightly worst score. One particular architecture (MLP with layers 54-500-800-7) was reported to achieve 68%; however, we didn't find this architecture to yield a better cumulative reward. Similarly, for the Adult data set, linear and deep classifiers were reported to achieve similar results (link) (around 84%), which is again equivalent to our cumulative reward of 4000 out of 5000. A specific DNN

| Name | d | A | Full memory | | Limited memory, Neural-Linear | | |
|---|---|---|---|---|---|---|---|
| | | | Linear (1) | Neural-Linear (2) | Both Priors (3) | No prior (4) | NeuralUCB (5) |
| Linear Datasets | | | | | | | |
| Mushroom | 117 | 2 | **11022 ± 774** | 10880 ± 853 | 11030 ± 810 | 7613 ± 1670 | -102 ± 84 |
| Jester | 32 | 8 | **14080 ± 2240** | 12819± 2135 | 11880 ± 283 | 11114 ± 2050 | 5373 ± 7 |
| Adult | 88 | 2 | **4066.1 ± 11.03** | 4010.0 ± 22.19 | 4027± 27 | 3608 ± 34 | 3751± 6 |
| Covertype | 54 | 7 | **3054 ± 557** | 2898 ± 545 | 2622 ± 166 | 2334 ± 603 | 1840.0 ± 34.62 |
| Nonlinear Datasets | | | | | | | |
| Census | 377 | 9 | 1791.5 ± 39.47 | 2135.5 ± 51.47 | **2733.6 ± 76.15** | 1195 ± 67 | 2103.2 ± 12.76 |
| Statlog | 9 | 7 | 4483 ± 353 | 4781 ± 274 | **4797.7 ± 48.21** | 3416 ± 42 | 4190 ± 13 |
| Epileptic | 178 | 5 | 1202.9 ± 34.68 | **1706.9 ± 41.26** | 1458.3 ± 76 | 1411 ± 33.43 | 1011 ± 2 |
| Smartphones | 561 | 6 | 3085.8 ± 24.64 | 3643.5 ± 64.89 | **4328.5 ± 87.75** | 1117 ± 28 | 2733 ± 713 |
| Scania Trucks | 170 | 2 | 4691.8 ± 7.23 | 4784 ± 6 | 4817.5 ± 541.89 | 4470.4 ± 37 | **4919.8 ± 193** |

Table 1: Cumulative reward of TS algorithms on 10 real world data sets. The context dim $d$ and the size of the action space $A$ are reported for each data set. The mean result and standard deviation of each algorithm is reported for $50$ runs.

was reported to achieve $90\%$ test accuracy but did not yield improvement in cumulative reward. These observations can be explained by the different loss functions that we optimize or by the partial observably of the bandit problem (bandit feedback). Alternatively, competitions tend to suffer from overfitting in model selection (see the "reusable holdout" paper for more details (Dwork et al., 2015)). Regret, on the other hand, is less prone to model overfitting because it is evaluated at each iteration and shuffles the data at each run.

Inspecting Table 5, we can also see that on all datasets using our memory limited algorithm with prior computations (Algorithm 3), improved the performance of the limited memory Neural-Linear, in which the priors are not updated (Algorithm 4). Furthermore, on six datasets (Mushroom, Adult, Census, Statlog, Smartphones, Scania Trucks), Algorithm 3 even outperformed the unlimited Neural-Linear algorithm (Algorithm 2). We can also see that in five (out of five) of the nonlinear data sets, the limited memory TS (Algorithm 3) outperformed Linear TS (Algorithm 1, which online as well). Our findings suggest that when the data is not linear, neural-linear bandits beat the linear method, even if they must perform with limited memory. In this case, computing priors improve performance and make the algorithm resilient to catastrophic forgetting.

We also compare our algorithm against NeuralUCB (Algorithm 5 Zhou et al. (2019)) with limited memory based on their offical code provided here . To make a fair comparison, we used the same network architecture. Also, NeuralUCB perform multiple network training iteration each step, while our online algorithm performs only one followed by a SGD step. We can see that NeuralUCB is outperformed by our algorithm on almost all datasets (except one) and poorly perform on the linear datasets. We attribute this results to the fact that NeuralUCB is based on NTK assumptions, which are not necessarily holds. Our likelihood matching technique can be seen as a fix between the NTK theory and applying it in practical problems.

## 5 RATE CONTROL FOR VIDEO TRANSMISSION OVER CELLULAR LINKS

The goal in this application is to control the sending rate of data segments sent over a cellular link, where the frequency of the changes to the sending rate should be low. In uplink video transmission the video encoder compresses the raw video into a stream of data to be transported. The encoder could adapt the video quality for near future throughput predictions, without inducing additional latency into the system. In this section, we show that both representation and exploration are advantageous on a simulated cellular link, as well as showing resilience to catastrophic forgetting.

We model the transmission rate control as a contextual bandit problem. At step $t$, a context $b(t) \in \mathbb{R}^d$ is revealed, which represents the current state of the link, and an action $a(t) \in A$ that controls the sending rate over the link is chosen by a policy. The action takes place for a fixed duration $\rho$ (seconds), and a reward $r_t$ is presented afterward. Let $(x_{i,t})_{i=1}^{k(t)}$ be the sequence of acknowledged (ACKed) packets observed during step $t$, where $k(t)$ is the number of ACKed packets during that step. Let $\sigma(x)$ be the size (in MB) of packet $x$, and $\delta(x)$ be the latency of packet $x$ in ms.

**Context Vectors**: Each context $b(t)$ is a vector that contains information about the state of the link during the last $h$ steps. After each step, we compute a vector of three features $u(t)$ that should

represents the state of the link at time $t$. Then, we construct the context $b(t + 1)$ by concatenating $(u(\tau))_{\tau=t-h+1}^{t}$. Specifically, for each $u(t)$ we measure the mean and variance of the latency values of the packets, and an approximation of the round-trip-time (RTT) by storing the minimal latency value observed over the last 100 steps. The values are normalized to keep the contexts bounded. The first context is generated by running the simulation for one step with a predefined initial action and using $u(t) = 0 \in \mathbb{R}^3$, $\forall t < 0$. **Actions**: The set of actions (arms) $A = \{a_i\}_{i=1}^{N}$, $N = 70$ is a set of evenly distributed values in the interval $[\lambda_{low}, \lambda_{high}] \subset \mathbb{R}$ such that $a_1 = \lambda_{low}$, $a_N = \lambda_{high}$, and $\forall 1 \leq i < N : a_i < a_{i+1}$. Each action value corresponds to a sending rate (in MBps).

**Rewards**: The reward signal is the total amount of data (in MB) that was transferred over the link with latency under $\eta$ ms minus the total amount of data with latency over $\eta$ ms during step $t$. Formally, $r_t = \sum_{i=1}^{k(t)} \sigma(x_{i,t}) \cdot g_\eta(x_{i,t})$, where $g_\eta(x_{i,t}) = 1$ when $\delta(x_{i,t}) < \eta$, and $-1$ otherwise.

**Cellular Link Simulation**: Many existing network simulators require significant resources (compute, time, human knowledge, etc.) to configure even just a single simulation setup, limiting the ability to obtain a diverse set of links quickly. We developed a queue-based simulation that requires only a few parameters and can model links with a wide range of behaviors. The simulation was designed to empirically fit patterns observed in data collected from uplink video transmission applications operating over real-world LTE networks. To make our agents more robust, during the training, we randomly sample links every $c$ steps given some boundary constraints on the parameters domain.

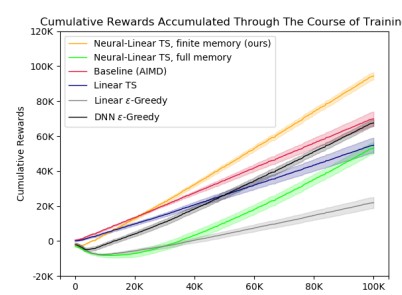

Figure 2: Cumulative rewards of the various algorithms. The results are averaged over 15 seeds. Shaded area corresponds to a 95% Gaussian confidence interval.

We compared our algorithm to the following baselines: linear TS, linear $\epsilon$-greedy, DNN $\epsilon$-greedy, unlimited memory neural-linear TS and a variant of the Additive-Increase-Multiplicative-Decrease (AIMD) control scheme, which is one of the dominant algorithms for congestion control. In our version of the algorithm, we control the sending rate at each step instead of controlling a congestion window size. Our AIMD will increase the sending rate by 0.1 MBps if the mean value of the latency was under RTT+50ms, otherwise, the sending rate would be cut in half. We used the same DNN architecture for both the neural linear and DNN $\epsilon$-greedy bandits. The architecture consists of two fully connected hidden layers with ReLU activations and sizes 128 and 32, respectively. For our algorithm, we used a memory buffer of size 2048. The other hyperparameters can be found in the supplementary material (Table 4).

Fig. 2 shows that the neural linear bandit benefits from both the nonlinear representations of the DNN, which allow it to improve over the linear TS bandit and from the efficient exploration, which allow it to improve over the DNN $\epsilon$-greedy bandit. Both DNN bandits improved over their linear counterparts, which shows that nonlinear approximations yield better predictions in this problem. Also, both TS bandits improved over their $\epsilon$-greedy counterparts, due to their efficient exploration mechanisms. It can also be seen that the AIMD baseline was competitive and surpassed all but our neural linear bandit in terms of cumulative rewards

# 6 SUMMARY

We presented a neural-linear contextual bandit algorithm that is resilient to catastrophic forgetting and demonstrated its performance on several real-world data sets. Our algorithm showed comparable results to previous methods that store all the data in a replay buffer. The algorithm approximately solves an SDP using SGD, which enables it to efficiently operates online. Our algorithm demonstrated excellent performance on multiple real-world data sets and especially on a challenging uplink transmission control problem. Moreover, its performance did not deteriorate due to the changes in the representation and limited memory. We believe that our findings make an important step towards solving contextual bandit problems where both exploration and representation learning play important roles. A main avenue for future work is to extend the ideas presented in this paper to Bayesian RL , where the immediate reward is replaced by the return, perhaps focusing on Markov decision processes with fast mixing time. An interesting future work would be to examine the effect of the network architecture on the performance of contextual bandits DNN-based algorithms such as ours and perhaps consider ways to choose the DNN architecture for contextual bandits problems as opposed to this work where the network's architecture is built for supervised learning, which in general may not be optimal for bandit problems.

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

## 7 PSEUDO CODE

---

**Algorithm 2** Limited Memory Neural-linear TS

---

Set $\forall i \in [1,..,N] : \Phi_i^0 = I_d, \hat{\mu}_i = \mu_i^0 = 0_d, \Phi_i = 0_{dxd}, f_i = 0_d$
Initialize Replay Buffer $E$, and DNN $D$
Define $\phi(t) \leftarrow \text{LastLayerActivations}(D(b(t)))$
**for** $t = 1, 2, \ldots,$ **do**
  **Observe** $b(t)$, evaluate $\phi(t)$
  **Posterior sampling:** $\forall i \in [1,..,N]$, sample:
    $\tilde{\mu}_i(t) \sim N\left(\hat{\mu}_i, \nu^2(\Phi_i^0 + \Phi_i)^{-1}\right)$
  **Play** arm $a(t) := \arg\max_i \phi(t)^T \tilde{\mu}_i(t)$
  **Observe** reward $r_t$
  **Store** $\{b(t), a(t), r_t\}$ in $E$
  **if** $E$ is full **then**
    Remove the first tuple in $E$ with $a = a(t)$ (round robin)
  **end if**
  **Posterior update:**
  $\Phi_{a(t)} = \Phi_{a(t)} + \phi(t)\phi(t)^T, f_{a(t)} = f_{a(t)} + \phi(t)^T r_t$
  $\hat{\mu}_{a(t)} = (\Phi_{a(t)}^0 + \Phi_{a(t)})^{-1}\left(\Phi_{a(t)}^0 \mu_{a(t)}^0 + f_{a(t)}\right)$
  **Update phase:**
  **if** $(t \bmod L) = 0$ **then**
    **for** $P$ steps **do**
      **Sample** batch $\{b_j, a_j, r_j\}_{j=1}^B$ from $E$
      Compute old features $\{\phi_j^{old}\}_{j=1}^B$
      Optimize $\omega$ on $\nabla_\omega \mathcal{L}_{NN}$ (Eq.1) and compute new features $\{\phi_j\}_{j=1}^B$
      $(\Phi_i^0)^{-1} \leftarrow (\Phi_i^0 + \Phi_i)^{-1}$ {Initial guess for SGD}
      **for** $\forall i \in [1,..,N]$ **do**
        $e_i^{old} \leftarrow \{\phi_j^{old}|a_j = i\}, e_i \leftarrow \{\phi_j|a_j = i\}$
        $(\Phi_i^0)^{-1} \leftarrow \text{ProjectedGradientDecent}((\Phi_i^0)^{-1}, e_i^{old}, e_i, \alpha)$
      **end for**
    **end for**
    **update using buffer:**
    **for** $\forall i \in [1,..,N]$ **do**
      Use the current weights of the last layer of the DNN as a prior for $\mu_i^0$
      $\Phi_i = \sum_{j=1}^{n_i} \phi_j^i(\phi_j^i)^T, f_i = \sum_{j=1}^{n_i}(\phi_j^i)^T r_j.$
    **end for**
  **end if**
**end for**

---

**Algorithm 3** ProjectedGradientDecent

---

**Inputs:** $A -$ PSD matrix, $\mathcal{B}_{old}, \mathcal{B}, \alpha$
**for** $\phi_j^{old} \in \mathcal{B}_{old}$ and $\phi_j \in \mathcal{B}$ **do**
  **Gradient step:**
  $s_j^2 \leftarrow (\phi_j^{old})^T A \phi_j^{old}$
  $X_j \leftarrow \phi_j(\phi_j)^T$
  $A \leftarrow A - \alpha\nabla_A ||\text{Trace}(X_j^T A) - s_j^2||^2$
  **Projection step:**
  $A \leftarrow \text{EigenValueThresholding}(A)$
**end for**

---

---

**Algorithm 4** Naive Limited Memory Neural-linear TS

Set $\forall i \in [1, .., N] : \Phi_i^0 = I_d, \hat{\mu}_i = \mu_i^0 = 0_d, \Phi_i = 0_{dxd}, f_i = 0_d$
Initialize Replay Buffer $E$, and DNN $D$
Define $\phi(t) \leftarrow \text{LastLayerActivations}(D(b(t)))$
**for** $t = 1, 2, \ldots,$ **do**
  **Observe** $b(t)$, evaluate $\phi(t)$
  **Posterior sampling:** $\forall i \in [1, .., N]$, sample:
    $\tilde{\mu}_i(t) \sim N\left(\hat{\mu}_i, \nu^2(\Phi_i^0 + \Phi_i)^{-1}\right)$
  **Play** arm $a(t) := \arg\max_i \phi(t)^T \tilde{\mu}_i(t)$
  **Observe** reward $r_t$
  **Store** $\{b(t), a(t), r_t\}$ in $E$
  **if** $E$ is full **then**
    Remove the first tuple in $E$ with $a = a(t)$ (round robin)
  **end if**
  **Posterior update:**
  $\Phi_{a(t)} = \Phi_{a(t)} + \phi(t)\phi(t)^T, f_{a(t)} = f_{a(t)} + \phi(t)^T r_t$
  $\hat{\mu}_{a(t)} = (\Phi_{a(t)}^0 + \Phi_{a(t)})^{-1}\left(\Phi_{a(t)}^0 \mu_{a(t)}^0 + f_{a(t)}\right)$
  **if** $(t \mod L) = 0$ **then**
    **for** $\forall i \in [1, .., N]$ **do**
      Evaluate old features on the replay buffer: $E_{\phi^{old}}^i$
    **end for**
    **Train** DNN for $P$ steps
    **Compute priors** for new features:
    **for** $\forall i \in [1, .., N]$ **do**
      Evaluate new features on the replay buffer: $E_\phi^i$
      Solve for $\Phi_i^0$ using Eq. (3) with $E_\phi^i, E_{\phi^{old}}^i, \Phi_i^{old}$
      Use the current weights of the last layer of the DNN as a prior for $\mu_i^0$
      $\Phi_i = \sum_{j=1}^{n_i} \phi_j^i(\phi_j^i)^T, f_i = \sum_{j=1}^{n_i} (\phi_j^i)^T r_j.$
    **end for**
  **end if**
**end for**

---

## 8 PROOF FOR LEMMA 1

*Proof.* We note that $X_{j,i} = \phi_j^i(\phi_j^i)^T$ is a 1-rank matrix. The stochastic gradient with respect to $(\Phi_i^0)^{-1}$, computed from a mini-batch of size $B$ taken from $E_\phi^i$, of the loss term is $2\sum_{j=1}^B X_{j,i}(\text{Trace}(X_{j,i}^T(\Phi_i^0)^{-1}) - s_{j,i}^2)$. By marking $\gamma_{j,i} = 2(\text{Trace}(X_{j,i}^T(\Phi_i^0)^{-1}) - s_{j,i}^2)$, the gradient can be presented as a weighted sum of 1-rank matrices: $\sum_{j=1}^B \gamma_{j,i}X_{j,i}$. Therefore, the rank of the stochastic gradient is at most $min\{B, g\}$. □

## 9 SENTIMENT ANALYSIS FROM TEXT USING CNNS

This is an experiment on the "Amazon Reviews: Unlocked Mobile Phones" data set. This data set contains reviews of unlocked mobile phones sold on "Amazon.com". The goal is to find out the rating (1 to 5 stars) of each review using only the text itself. We use our model with a Convolutional Neural Network (CNN) that is suited to NLP tasks (Kim, 2014; Zahavy et al., 2018b). Specifically, the architecture is a shallow word-level CNN that was demonstrated to provide state-of-the-art results on a variety of classification tasks by using word embeddings, while not being sensitive to hyperparameters (Zhang & Wallace, 2015). We use the architecture with its default hyper-parameters (Github) and standard pre-processing (e.g., we use random embeddings of size 128, and we trim and pad each sentence to a length of 60). The only modification we made was to add a linear layer of size 50 to make the size of the last hidden layer consistent with previous experiments.

In this experiment, the input dimension is so large ($\mathbb{R}^{7k}$), so we could not run a linear baseline since it is impractical to do linear algebra (e.g., calculate an inverse) in this dimension. Instead, we compare the proposed method – neural-linear

| $\epsilon-$greedy | Neural-Linear | Neural-Linear Limited Memory |
|---|---|---|
| $2963.9 \pm 68.5$ | $3155.6 \pm 34.9$ | $3143.9 \pm 33.5$ |

Table 2: Cumulative reward on Amazon review's

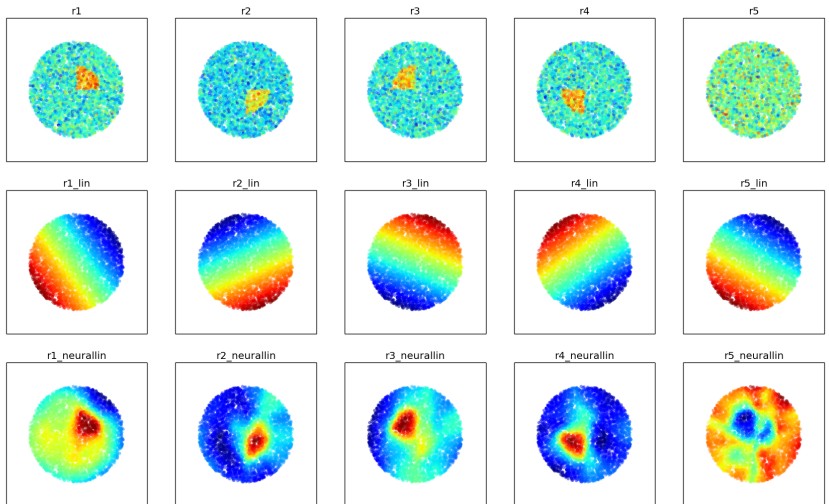

Figure 3: Representations learned on the wheel data set with $\delta = 0.5$. Reward samples (top), linear predictions (middle) and neural-linear predictions (bottom). Columns correspond to arms.

with finite memory – with the neural linear TS

baseline and with an additional baseline that uses an $\epsilon-$greedy exploration scheme (which is also not limited by memory). We experimented with 10 values of $\epsilon$, $\epsilon \in [0.1, 0.2, ..., 1]$ and report the results for the value that performed the best (0.1). Looking at Table 2, we can see that the limited memory version performs almost as good as the full memory, and better than the $\epsilon-$greedy baseline.

## 10 NON LINEAR REPRESENTATION LEARNING ON A SYNTHETIC DATA SET

**Setup:** we adapted a synthetic data set, known as the "wheel bandit" (Riquelme et al., 2018), to investigate the exploration properties of bandit algorithms when the reward is a nonlinear function of the context. Specifically, contexts $x \in \mathbb{R}^2$ are sampled uniformly at random in the unit circle, and there are $k = 5$ possible actions.

One action , $a_5$, always offers reward $r_5 \sim N(\mu_5, \sigma)$, independently of the context. The reward of the other actions depend on the context and a parameter $\delta$, that defines a $\delta-$circle $\|x\| \leq \delta$.

For contexts that are outside the circle, actions $a_1, .., a_4$ are equally distributed and sub-optimal, with $r_i \sim N(\mu, \sigma)$ for $\mu < \mu_5, i \in [1..4]$.

For contexts that are inside a circle, the reward of each action depends on the respective quadrant. Each action achieves $r_i \sim N(\mu_i, \sigma)$, where $\mu_5 < \mu_i = \dot{\mu}$ in exactly one quadrant, and $\mu_i = \mu < \mu_5$ in all the other quadrants. For example, $\mu_1 = \dot{\mu}$ in the first quadrant $\{x : \|x\| \leq \delta, x_1, x_2 > 0\}$ and $\mu_1 = \mu$ elsewhere. We set $\mu = 0.1, \mu_5 = 0.2, \dot{\mu} = 0.4, \sigma = 0.1$. Note that the probability of a context randomly falling in the high-reward region is proportional to $\delta$. For lower values of $\delta$, observing high rewards for arms $a_1, .., a_4$ becomes more scarce, and the role of the nonlinear representation is less significant.

We train our model on $n = 4000$ contexts, where we optimize the network every $L = 200$ steps for $P = 400$ mini batches. The results can be seen in Table 3.

Not surprisingly, the neural-linear approaches, even with limited memory, achieved better reward than the linear method (Table 3).

Fig. 3 presents the reward of each arm as a function of the context. In the top row, we can see empirical samples from the reward distribution. In the middle row, we see the predictions of the linear bandit. Since it is limited to linear predictions, the predictions become a function of the distance from the learned hyper-plane. This representation is not able to separate the data well, and also makes mistakes due to the distance from the hyperplane. For the neural linear method (bottom row), we can see that the DNN was able to learn good predictions successfully. Each of the first four arms

learns to make high predictions in the relevant quadrant of the inner circle, while arm 5 makes higher predictions in the outer circle.

|  | Linear | Neural-Linear Limited Memory |
|---|---|---|
| $\delta$=0.5 | $737.44 \pm 3.04$ | $899.72 \pm 12.79$ |
| $\delta$=0.3 | $735.37 \pm 2.58$ | $781.09 \pm 11.34$ |
| $\delta$=0.1 | $735.51 \pm 2.59$ | $751.75 \pm 3.6$ |

Table 3: Cumulative reward on the wheel bandit

## 11    PARAMETERS FOR THE TRANSMISSION RATE CONTROL EXPERIMENT

| Parameter | Value |
|---|---|
| $\eta$ | 120 |
| $c$ | 16 |
| $\rho$ | 2 |
| $N$ | 70 |
| $\lambda_{low}$ | 0.01 |
| $\lambda_{high}$ | 7 |
| $T$ | $10^5$ |
| $h$ | 5 |
| $P$ | 256 |
| $L$ | 512 |
| $\epsilon$ | 0.05 |
| Learning rate | 3e-4 |
| Memory buffer size | 2048 |

Table 4: Hyper-parameters values for the experiment in section 5

## 12    OLD RESULTS

|  |  |  | Full memory | | Limited memory, Neural-Linear | | |
|---|---|---|---|---|---|---|---|
| Name | d | A | Linear | Neural-Linear | Both Priors | $\mu$ Prior | No Prior |
| Linear Datasets | | | | | | | |
| Mushroom | 117 | 2 | $11022 \pm 774$ | $10880 \pm 853$ | $10923 \pm 839$ | $9442 \pm 1351$ | $7613 \pm 1670$ |
| Financial | 21 | 8 | $4588 \pm 587$ | $4389 \pm 584$ | $4597 \pm 597$ | $4311 \pm 598$ | $4225 \pm 594$ |
| Jester | 32 | 8 | $14080 \pm 2240$ | $12819 \pm 2135$ | $9624 \pm 2186$ | $10996 \pm 2013$ | $11114 \pm 2050$ |
| Adult | 88 | 2 | $4066.1 \pm 11.03$ | $4010.0 \pm 22.19$ | $3943.0 \pm 54.29$ | $3839.5 \pm 17.63$ | $3608.2 \pm 34.94$ |
| Covertype | 54 | 7 | $3054 \pm 557$ | $2898 \pm 545$ | $2828 \pm 593$ | $2347 \pm 615$ | $2334 \pm 603$ |
| Nonlinear Datasets | | | | | | | |
| Census | 377 | 9 | $1791.5 \pm 39.47$ | $2135.5 \pm 51.47$ | $2023.16 \pm 37.3$ | $1873 \pm 757$ | $1943.83 \pm 84.2$ |
| Statlog | 9 | 7 | $4483 \pm 353$ | $4781 \pm 274$ | $4825 \pm 305$ | $4681 \pm 285$ | $4623 \pm 276$ |
| Epileptic | 178 | 5 | $1202.9 \pm 34.68$ | $1706.9 \pm 41.26$ | $1716.8 \pm 60.44$ | $1572.9 \pm 48.66$ | $1411.0 \pm 33.43$ |
| Smartphones | 561 | 6 | $3085.8 \pm 24.64$ | $3643.5 \pm 64.89$ | $2660.4 \pm 84.72$ | $3064.5 \pm 55.06$ | $2851.6 \pm 58.77$ |
| Scania Trucks | 170 | 2 | $4691.8 \pm 7.23$ | $4784.7 \pm 6.05$ | $4742.0 \pm 33.0$ | $4698.0 \pm 13.06$ | $4470.4 \pm 37.9$ |

Table 5: Cumulative reward of TS algorithms on 10 real world data sets. The context dim $d$ and the size of the action space $A$ are reported for each data set. The mean result and standard deviation of each algorithm is reported for 50 runs.

