# OpenReview forum: "Online Limited Memory Neural-Linear Bandits"
_ICLR.cc/2021/Conference — Reject_

### Official Review · AnonReviewer1 · 2020-10-26
**The paper proposes the limited-memory neural-linear bandit algorithm. Neural-linear bandit extends the linear bandit algorithm by allowing a deep neural network for estimating the reward function from a given context. It uses the last layer in the DNN as the nonlinear representation and do Thompson sampling based exploration on the last linear layer. The paper proposes the technique to handle limited memory, avoiding saving a large amount of historical information in the memory.**

**Rating:** 5
**Confidence:** 3

**Review:**

The main contribution of the paper is the likelihood matching method for dealing with the catastrophic forgetting and save the memory usage. The empirical results demonstrate that the proposed method achieves good results comparing to other baselines.

However, the paper has several issues, as discussed below.

- The paper lacks the theoretical result on their algorithm. It is understandable that an algorithm based on DNN may be hard to analyze. But the NeuralUCB algorithm of Zhou et al. (2019) does provide a theoretical regret analysis, even though NeuralUCB is also based on a DNN. Therefore, at least some more discussions and comparisons are needed. What are the difficulties for the regret analysis comparing to NeuraUCB? In general, a more detailed comparison with NeuralUCB is needed, also see the empirical evaluation part below.

- The realizability assumption (Assumption 2) on the DNN is quite strong. It requires that ANY representation \phi produced by DNN is realizable, that is, have a corresponding unknown but fixed parameters \mu_i that could generate the reward of i with representation \phi(t) by a linear combination. The authors only mention that the assumption is too strong to hold in practice, but it does allow their algorithm to perform well in many problems. So what is the reason behind? is there any way to relax it and make it more reasonable? It is difficult to accept an approach that is based on an impossible assumption. I think this is linked to the underlying principle of the proposed approach. The DNN is part of the proposed algorithm that needs to be constructed and tuned from data, but when taking its last year for TS exploration, it seems that the authors are treating this entire DNN as given by the environment, as a transformation from the context b(t) to a representation \phi(t), and thus making an assumption patterned from the Assumption 1 for linear bandit, which is entirely on the environment. This mixing of algorithm and environment creates the difficulty of both justifying the assumption and carrying out the analysis I believe. I hope that the authors could provide more discussions on this important point. Is there any way to alleviate this, such as considering a class of DNNs so that such a realizability assumption could be supported?

- For the empirical comparison, the authors do not compare the NeuralUCB algorithm in Zhou et al. 2019. In fact, this paper already appears in ICML'2020. Since Thompson Sampling and UCB are two main approaches for stochastic MAB, it is important to compare the Neural-Linear algorithm of the current paper with NeuralUCB algorithm, both analytically and empirically. Moreover, I do not understand a result reported in Figure 2. In this figure, it shows that the Neural-linear TS with full memory performs much worse than Neural-Linear TS with finite memory. How could this be? To me the finite memory algorithm (proposed in the paper) is an approximation of the full memory version. It only gains in saving memory, but why could it also win in reward?

- The writing of the paper is not entirely clear. Some technical parts of the paper is not easy to follow. For example, at the beginning of Page 4, it starts to use notations \Phi_i and \Phi^0_i, but they are not defined. Then \Phi_i appears in Eq.(2), seemingly to be a definition, but \Phi^0_i is still not defined. It turned out that at the end of this page, \Phi^0_i is discussed as the correlation matrix connecting the new features with the old ones. But it still does not look like a solid technical definition to me. Without a clear definition up front on these key notations, it is very hard to understand the entire logical flow. Also, in line 3 of Page 4 mentions the noise parameter \nu in Alg. 1, but there is no such parameter in the Alg. 1 pseudocode. I am guessing it is the v in the pseudocode. The authors need to provide rigorous technical definitions and the logical flow to help readers understand their approach.

---

### Official Review · AnonReviewer3 · 2020-10-26
**Review Online Limited Memory Neural-Linear Bandits**

**Rating:** 5
**Confidence:** 3

**Review:**

Online Limited Memory Neural-Linear Bandits
The presented paper suggests a method for neural linear bandits that use limited memory. The memory limit is introduced by forgetting past observations and fitting the likelihood of the data under moments constraints using a semi definite program. The paper presents an interesting approach to a relevant problem but lacks novelty. Moreover, the presentation of the paper needs improvement (i.e. figures are not readable and results should be highlighted better in the tables). The paper needs another pass to correct writing mistakes and errors in the mathematical formulas.

Introduction:
Representation change -> representation changes
The are -> there are
What do you mean by computational problems?
Please change the citation style (no parentheses)
What do you mean by “patch-based”?


Background:
First equation: what do you mean by \sim? Is this rather \propto?
Plays the arm that maximises -> use \tilde{\mu} here

Limited memory NL TS
The part on exploration is not clear, please improve readability
Inv-Gamma(a,b) you use a and b here (like action and context), please change notation

Experiments:
Plot is not readable
Table would be better if best were highlighted in bold

Rate Control:
Plot not readable

---

### Official Review · AnonReviewer2 · 2020-11-02
**A challenging problem but a simple solution without theoretical guarantees**

**Rating:** 3
**Confidence:** 4

**Review:**

The paper proposed a neural-bandit approach using Thomson sampling for leveraging the DNN’s non-linear representation. There is a growing interest in using Multi-armed Bandits with DNN in an end-to-end learning fashion but the difficulty of the understanding how the representation is learned in DNN, makes this problem a very challenging and interesting one. The paper proposes one solution to consider explore-exploit tradeoff in the non-linear DNN space. There is still several unanswered questions in this paper and leave the readers with more confusion than clarify. This is not because the paper is not written properly but the topic is fairly new and a very few works have considered this problem setting. The likelihood matching seems interesting idea but I couldn’t find a reason whether it contributed to reward boost compared to using all the history. Some of the claims in the paper is demonstrated experimentally, not theoretically. Please clarify the following questions to understand the paper better.

### Major Questions:

* The realizability assumption A2 doesn’t seems to apply here as the history of the decisions (stored in the replay buffer) is used to update the representation.

* The paper says $\mu$ is assumed to be fixed in this problem, even before the representation is known. As we update the representation,$\mu$ for the new representation varies and cannot be fixed.

* Memory is poorly used without any significance to past decisions that are somehow key to changing the reward distribution or representation or both.

* Unlike in the earlier work (Zhou et. Al), no theoretical guarantees have been provided. As in the earlier work, Network width/depth based guarantees can be used to show how the representation affects the realization assumption.

* It is unclear why Alg 3 with limited memory outperforms Alg 2 with Full memory (stores all the past decisions) on both linear and non-linear datasets (Mushroom, Financial, Statlog, Epileptic). Please clarify.

### Minor Questions:

* $\Phi_0$ was introduced before explaining about the likelihood matching. I suggest moving the updates at the end.

* It would be beneficial for the paper if an additional experiment on how varying the size of the memory buffer affect the cumulative rewards

* The paper claims that the linear models work only for ”medium-sized” inputs (with around 1000 features) due to numerical issues. Can you provide any further analysis on this? Perhaps varying the dim d vs cumulative rewards for linear vs Neural linear would demonstrate this claim. If there is an earlier work that did this, the citation would suffice.

I have read the author's comments and I stand with my previous rating. I believe the paper addresses an interesting problem but lacks sufficient analysis due to the realizability assumption A2 which doesn't apply for the given problem and the other reviewers feel the same way. Unlike in online learning and MAB, the memory used by the proposed Neural-linear bandit significantly deviates from the A2 assumption. These should have been analyzed empirically or theoretically to understand the impact of the past history on the regret.

---

### Author Response · Authors · 2020-11-22
**General comment for all of the reviewers**

We would like to thank the reviewers for their constructive feedback. Below we address the main concerns that were raised by the reviewers:

1. Comparison to NeuralUCB (Zhou et al.): Our work is not an alternative to the NeuralUCB algorithm but a complementary idea. Our focus is on the representation shift that the agent occurs during learning (under memory limitations). The theoretical analysis in Zhou et al. is a great contribution to the field, but it addresses the regret of the algorithm under the NTK assumptions and does not address this phenomena. We will add an empirical comparison of our approach to the  NeuralUCB method.

2. The realizability assumption (A2): As mentioned in the paper, this assumption is indeed strong and we are not sure if it will hold in the real world. That said, analyzing the problem under this assumption is what motivated us to design an algorithm that performs well on real-world data sets. This is also true for the standard realizability assumption: it motivated many interesting and well analyzed algorithms that work well in practice, while it's unclear if the assumption itself holds with real data.

3. Full memory neural-linear TS results - We are aware that in some of the experiments our memory-limited approach achieves better results against the full-memory version. This can be explained by the fact that sometimes you may benefit from forgetting as your past decisions may lead to the use of a sub-optimal policy.

4. Ablation experiment of memory size -  great idea, will add them to the main paper.

We will rewrite the paper in order to add/fix these issues and other notes that the reviewers mentioned.

We want to thank you again for the informative review, the authors.

---

### Author Response · Authors · 2020-11-24
**Updated paper**

We compared our algorithm against NeuralUCB, provided in Table 1 in the updated paper. We used the code provided by the authors in https://github.com/ZeroWeight/NeuralTS.
At the updated table we compare our online version algorithm against NeuralUCB while the old results are provided at the supplementary. Please note that columns of algorithms (3) and (4) in the new table correspond to the online version.
We also update the paper according to the helpful rejects of the reviewers that could be handled at the time frame of the rebuttal.

---

### Decision · Program_Chairs · 2021-01-07
**Final Decision**

**Decision:**

Reject

**Comment:**

This paper proposes a promising solution to a very interesting and challenging problem, and the authors have improved the paper during the rebuttal by adding an important missing baseline. However, all reviewers still agree that the paper currently lacks sufficient analysis that would be required to understand properly the implications of past history on the regret. More specifically, the fact that assumption A2 does not apply to the given problem raises questions that should be addressed before publication. Theoretical analysis was provided for previous similar work (e.g. NeuralUCB). Providing this for the proposed method would significantly improve the impact of this work.